# IoT Based Smart Monitoring of Patients’ with Acute Heart Failure

**DOI:** 10.3390/s22072431

**Published:** 2022-03-22

**Authors:** Muhammad Umer, Saima Sadiq, Hanen Karamti, Walid Karamti, Rizwan Majeed, Michele NAPPI

**Affiliations:** 1Department of Computer Science, Khwaja Fareed University of Engineering and Information Technology, Rahim Yar Khan 64200, Pakistan; umer.sabir@iub.edu.pk (M.U.); cosc201701005@kfueit.edu.pk (S.S.); 2Department of Computer Science Information Technology, The Islamia University of Bahawalpur, Bahawalpur 63100, Pakistan; 3Department of Computer Sciences, College of Computer and Information Sciences, Princess Nourah bint Abdulrahman University, P.O. Box 84428, Riyadh 11671, Saudi Arabia; hmkaramti@pnu.edu.sa; 4Department of Computer Science, College of Computer, Qassim University, Buraydah 51452, Saudi Arabia; w.karamti@qu.edu.sa; 5Data Engineering and Semantics Research Unit, Faculty of Sciences of Sfax, University of Sfax, Sfax 3052, Tunisia; 6Directorate of Information Technology, The Islamia University of Bahawalpur, Bahawalpur 63100, Pakistan; rizinbox@gmail.com; 7Department of Computer Science, University of Salerno, 84084 Fisciano, Italy

**Keywords:** IoT, smart healthcare, patient’s mortality prediction, deep learning, heart disease

## Abstract

The prediction of heart failure survivors is a challenging task and helps medical professionals to make the right decisions about patients. Expertise and experience of medical professionals are required to care for heart failure patients. Machine Learning models can help with understanding symptoms of cardiac disease. However, manual feature engineering is challenging and requires expertise to select the appropriate technique. This study proposes a smart healthcare framework using the Internet-of-Things (IoT) and cloud technologies that improve heart failure patients’ survival prediction without considering manual feature engineering. The smart IoT-based framework monitors patients on the basis of real-time data and provides timely, effective, and quality healthcare services to heart failure patients. The proposed model also investigates deep learning models in classifying heart failure patients as alive or deceased. The framework employs IoT-based sensors to obtain signals and send them to the cloud web server for processing. These signals are further processed by deep learning models to determine the state of patients. Patients’ health records and processing results are shared with a medical professional who will provide emergency help if required. The dataset used in this study contains 13 features and was attained from the UCI repository known as Heart Failure Clinical Records. The experimental results revealed that the CNN model is superior to other deep learning and machine learning models with a 0.9289 accuracy value.

## 1. Introduction

Internet of things (IoT) based embedded systems are now transforming into interconnected smart devices using sensors. The main challenge of using smart devices was low storage capacity and limited processing power, which was solved by cloud computing by providing high-level processing and storage capacities. There is a need for the incorporation of IoT sensors with cloud computing to deal with smart setup for healthcare [1]. In the smart healthcare system, patients are being monitored in real-time. A smart framework is essential for monitoring patients and for providing high-quality and quick healthcare at a low cost using deep learning and artificial intelligence (AI). AI, along with the IoT-cloud framework, have shown advancements in providing real time based smart healthcare system and improved decision making ability. Smart mobile devices can be linked to sensors and can help with providing a quick response of medical professionals in a smart healthcare setup. Patients in intensive care need a quick response, proper monitoring and immediate care to stay alive. It is difficult for medical experts to be available in hospital at all times. Therefore, a smart framework is required to resolve these issues in healthcare systems.

The use of IoT sensors with cloud computing is an emerging research area. Low cost and advanced IoT sensors are being used in smart healthcare systems. These sensors include wearable devices to monitor diabetes, temperature, blood pressure, anemia, sodium level, creatinine, and ECG. Real-time data obtained by the smart sensors of IoT devices are complex to handle and require efficient methods such as big data analytics, cloud computing, AI, and deep learning. These latest technologies provide facilitation in data processing, analysis, and storage. A big challenge for a smart health care system where various IoT sensors produce a large amount of data is a need for a healthcare framework that could benefit all types of stakeholders by providing cost-effective and efficient solutions.

All technological advancements in IoT-based devices cannot deal with the complexities of the data handling of any smart framework without having cognitive ability. Many researchers have applied cognitive ability to developing smart IoT-based systems. In the scenario of an IoT-based smart healthcare system, all healthcare indicators are considered to determine whether an emergency service is required by informing healthcare professionals. The authors used IoT-based devices for smart healthcare systems using cloud computing to obtain patients’ records and to monitor them [2]. A cloud-based IoT framework is proposed by many researchers such as for emotion recognition using deep learning [3], healthcare systems for cities [4], smart city using CNN [5], and healthcare systems to monitor COVID-19 patients [6]. All these efforts have been performed to improve the smart healthcare system.

As stated by the World Health Organization (WHO), cardiac diseases are the foremost cause of death globally with 17.9 million deaths annually [7]. High cholesterol, hypertension, obesity, and hyperglycemia all increase the risk of heart problems [7]. Sleep disturbance, chronic cough, swelling in legs and fast heart rate are symptoms related to the heart [8]. Diagnosis of heart disease is challenging in its early stages for health professionals, as the nature of these symptoms can be confused with problems of aging. Angiography is being used by professionals for accurate diagnosis of cardiac artery disease [9], which is very expensive and difficult to afford for impoverished people. Growing medical data provide suitable circumstances for improving disease diagnosis. Advancement in computer technologies helps professionals with their decision-making. Machine learning (ML) is an effective tool for converting a large collection of medical data into useful knowledge [10].

Data mining is applied to extract hidden useful information from large past data collection for future decision-making. It is being used in different fields such as education, medicine and engineering. ML-based algorithms have been utilized to deal with complex and non-linear data by improving prediction outcomes [11]. These techniques need to be explored more to facilitate medical professionals in diagnostic decisions. Different ML-based models are employed to predict heart disease severity and death in heart patients [12]. A dataset containing records of heart failure patients was developed by [13]. Data were collected from the Institute of Cardiology and the Allied Hospital Faisalabad, Pakistan. They made the dataset publicly available for further exploration. Afterwards, the same dataset was utilized by researchers and they predicted mortality rate [14]. After that, two features of the same dataset were utilized by [15] in predicting the performance of ML-based models. Eventually, the same dataset was explored by [16], using the full set of features in comparing ML-based models to predict heart failure survival. The authors also applied the Synthetic Minority Oversampling Technique (SMOTE) to make the dataset balanced. Although the results obtained by ML-based methods are reasonable, there is still room for improvement in dealing with large-sized high dimensional datasets. Therefore, there is a need to explore deep learning models to predict heart failure survivals, as these techniques have shown robust results on large datasets.

Pathology reports of patients have been widely utilized by researchers for smart healthcare systems. Medical records of patients require expert knowledge and time for analysis. Smart Healthcare systems need proper training to perform well. For this purpose, a smart healthcare system uses machine learning or deep learning models to make predictions or decisions. In this study, the main goal is to propose a smart healthcare framework that is based on patients’ pathology reports. The proposed model also investigates deep learning models in classifying heart failure patient as alive or deceased. The framework employs IoT-based sensors to obtain signals and send them to the cloud web server for processing. These signals are further processed by deep learning models to determine the state of patients and inform medical staff. Then medical staff will decide to take action accordingly and will generate an emergency response if needed. Major contributions of this study are:Designed an efficient smart healthcare system based on IoT and cloud-based technologies to provide a timely health care service to heart-failure patients using a deep learning model;This is the first study to design a smart healthcare system to monitor heart failure patients using the Heart-failure-clinical-records-dataset;Performance of deep learning models is investigated in predicting the survival of heart patients;The performance of the proposed CNN deep learning model is compared with MLP, RNN, LSTM and ML-based algorithms trained on the same dataset.

The rest of the paper is organised as follows: Section 2 discusses previous literature. Section 3 provides the scenario of the proposed smart framework. Section 4 describes the dataset used in this work. It also explains the various algorithms used in this research. Section 5 explains experimental details and evaluation parameters. Section 6 discusses the detailed discussion of the result. The conclusion and future work are given in Section 7.

## 2. Related Work

Smart healthcare systems are receiving attention nowadays and are also providing socio-economical benefits in designing smart cities. IoT-based sensors and cloud computing combined have revolutionized the concept of a smart system. Smart healthcare systems include the remote monitoring of patients, the detection of disease, providing healthcare remotely, smart equipment, telemedicine, and remote medical operations. Such a system provides quick medical facilities in case of any emergency. In this setup, sensors are attached to the patient’s body and provide real-time data on the patients. Researchers designed a framework to handle the electronic records of the patients [17]. To monitor a patient’s glucose level, the authors propose a smart healthcare system [18]. A smart ambulance controlled by robots was designed to provide an emergency health service to cardiovascular patients [19]. A data secure management system over the 6G network is proposed in [20]. Some studies have been performed to detect a forgery in the smart setup of healthcare systems [21]. The authors discussed protocols and challenges of IoT [22].

In recent times, ML has been applied to predict different cardiovascular diseases. Melillo et al. [23] proposed an automated system to separate high-risk heart failure patients from low-risk patients. They applied the Classification and Regression Tree (CART) and achieved a specificity value equal to 63% and sensitivity value equal to 93%. Guidi et al. [24] designed a decision support system to analyze heart failure patients. Authors compared different classifiers such as Support Vector Machine (SVM), Random Forest (RF), fuzzy rule-based CART, and Neural Network (NN). The RF and CART outperformed with 87% accuracy. Parthiban et al. [25] utilized SVM for the diagnosis of cardiac disease in diabetic patients and achieved 94% accuracy. Electronic health records are very useful for research and clinical purposes [26]. Minor errors in physical examinations of heart patients can be a danger for a patient’s life while machine learning-based models have been effectively diagnosing heart diseases and reducing the death ratio [27].

Shah et al. [28] investigated different health conditions that are primary factors of death and causing heart disease. Different ML-based models, such as Naive Bayes (NB), Decision Tree (DT), Random Forest (RF) and K Nearest Neighbour (KNN), were applied by the authors. They applied the feature selection technique and used 14 out of 76 features to meet their required goal. Many researchers applied ensemble approaches to improve the prediction results. A hybrid model in combination with a novel feature representation method to predict heart disease was proposed by [29]. An optimized ML model SVM was designed by [30] that used CG signals to predict heart disease. Manogaran et al. [31] applied an Adaptive Neuro-Fuzzy Inference System with Multiple Kernel Learning to diagnose heart disease and achieved good results.

A major challenge of ML-based models is large-scale and high-dimensional datasets [32]. The analysis of numerous features requires a large memory space and it can lead to the over-fitting of the model [33]. Therefore assigning different weights help to decrease processing time and also improve the model’s performance [34,35]. The identification of a subset of features facilitates disease diagnosis, health management, and IoT. The dimensionality reduction of features is performed by extracting features first and then removing useless features from the dataset [36]. Heart-related features were reduced by Generalized Discriminant Analysis, Extreme Learning Machine, and ranking method and the accuracy of heart disease prediction was improved [37]. Authors also used Generalized Discriminant Analysis to reduce features with Multilayer perceptron (MLP) and improved classification results [38]. Mohammadzadeh et al. [39] reduced 15 features to five features of heart rate variability with SVM and achieved 100% precision. Authors proposed a wearable ECG sensor and also proposed an energy efficient algorithm which saves 35% of energy during sensing and transmission [40]. Researchers applied an ensemble model to predict heart disease by using sensor data in a fog computing environment [41].

Recent studies demonstrated that feature selection techniques improved the prediction accuracy of cardiac problems. Al Rahhal et al. [42] used a neural network model for ECG signals classification to analyze the top features. Researchers involved expert opinion at each interval during the training phase which can create bias. An ensemble stacked model was proposed by [43], which combined RF with Gradient Boosting Machine (GBM) and Extreme Gradient Boosting (XGBoost). The authors also applied an optimized feature selection technique on four datasets. To show the robustness of their proposed model, they performed 10-fold cross-validation. Another ensemble of neural network models was proposed by [44]. The authors employed CNN with BiLSTM and BiGRU and achieved robust results. They also applied the feature selection technique and several preprocessing steps.

An extensive study of the literature revealed that many attempts have been made by researchers in predicting cardiac diseases. Different optimization methods, such as appropriate feature representation techniques in combination with feature selection methods, have been employed on different datasets. These techniques performed well in the classification of cardiovascular diseases. On the basis of discussion, existing work is summarized in Table 1. Analysis revealed that existing approaches suffer from shortcomings. Firstly, many researchers have used small-sized datasets which makes classification results not reliable or general enough. Secondly, various researchers have applied an ensemble of ML models that are built on weak individual models so the desired results have not been achieved. Research based on deep learning models are mostly applied on image-based datasets, ECG signal-based datasets, or $Spo_{2}$ signal-based datasets. These types of data required a specialized equipment for data collection. Data collection using smart sensors is an easy and fast way. However, to deal with these limitations, the main aim of this study is to design a smart healthcare system based on a deep learning model to monitor heart failure patients. The Heart-failure-clinical-records-dataset has been achieved by the UCI repository to validate the proposed model. Consequently, the performance results achieved by the proposed model surpassed the existing literature.

## 3. Smart Healthcare Framework

This section presents an IoT and cloud-based proposed healthcare framework for Heart Failure Patients. The architecture of the proposed smart healthcare system is presented in Figure 1. This framework is designed to facilitate medical professionals in monitoring patients’ health using IoT-based devices. Medical staff can access medical records of heart failure patients any time and anywhere using IoT and cloud-based technologies. To monitor the healthcare condition of heart patients, sensors are used to monitor Heart Rate (HR), Blood Pressure (BP), Temperature, Blood Glucose and Cholesterol and Electrocardiogram signals (ECG) [45]. The parameters provide help in monitoring general health status of the heart patients. Smart wearables and fixed sensors enable transmitting patient data and update it using IoT as shown in Figure 2. Sensors are easy to handle for elderly and weak patients. The selection of devices is based on the hospital and care center facility. The proposed framework analyzes real-time information of the patients and helps patients to get medical emergency services in a short time. Patient-related records are uploaded on the cloud, therefore, it is available for the medical staff remotely and they can provide advice according to the state of the patient.

The main aim of the proposed smart healthcare framework is to provide easy, low cost and reliable monitoring of heart failure patients and improve the chance of survival of critical patients. The proposed system gets signals from the sensors and transmits these signals to the cloud for further processing by applying deep learning models. The deep learning model then predicts alive or deceased patients on the basis of the data accessed through sensor signals. Based on these results medical professionals plan their further activities to deal with patients.

Data arranged in the form of a patient’s diagnostic information are shared with medical experts for detailed analysis. Medical professionals have smart devices such as smartphones, digital assistants, or tablets. These devices can store data locally and can perform simple computations. Then medical professionals perform follow up with heart failure patients accordingly. The flow of the proposed IoT framework is presented in Algorithm 1.
**Algorithm 1** The steps of the proposed IoT architecture based on the Deep Learning Model.**Read:** The smart medical healthcare sensors data.
**Connect:** Make connection to firebase database for transferring data.
**Authentication:** Medical officer authentication.
**IF:** Transfer==’successful’
**1.** Transferring of smart medical healthcare sensors data using JSON dump method.
**2.** Already trained deep learning model make predictions on sensor data.
**3.** Based on predictions medical report of patient is generated with remarks as prediction.
**4.** Report is sent back from firebase cloud storage to medical officer device.
**Else:** Transfer==’Unsuccessful’
**1.** Smart medical healthcare sensors data is stored in the shared preference of the device.
**2.** Shared preference data is sent to firebase cloud storage whenever the connectivity is successful.


## 4. Materials and Methods

### 4.1. Dataset

In this study, the Heart-failure-clinical-records-dataset [13] is obtained from the UCI ML repository [46]. The dataset consists of patients’ medical history records having heart issues. Data were collected during the follow-up time period and contain 13 clinical features. From a total of 299 patient’s records, 194 records are male, 105 records are female. The age of all patients is more than forty years. One represents a deceased patient and zero represents an alive patient in the target class. All the patients suffered from left ventricular systolic dysfunction and had heart failure problems already. The detail of the dataset is presented in Table 2.

### 4.2. Deep Leaning Models

Deep learning is an emerging research area in the domain of Artificial Intelligence (AI). It provides promising results by end-to-end modeling of data. The use of an automated system by medical experts in disease diagnosis has proved to be a very effective and functional tool. Deep learning is popular for dealing with a colossal amount of data. Currently, it is extensively being used in medical data analysis as it does not require manual feature extraction.

#### 4.2.1. Multilayer Perceptron Neural Network

MLP has dominant characteristics with respect to classification, such as it is fast, easy to implement, and requires small sized training sets [47]. MLP has three main layers which are the input layer, output layer and hidden layers. Hidden layers are middle layers that connect input layer to output layer after processing. In hidden layers of MLP, *j* is the sum of input values after multiplication with its respective weights $w_{ij}$ and output $y_{j}$ is computed as sum. Mathematically, it is presented as:$$y_{j}=f\left(\sum w_{ij}*O_{i}\right),$$
where *w* represents weight, which is assigned by a gradient descent algorithm and *O* represents hidden layers.

Gradient descent is an efficient algorithm and is used to train neural networks [48]. It updates randomly selected weights to a negative gradient after calculating derivative of the error function. The training of the algorithm is stopped after reaching maximum iterations. The gradient descent algorithm is used to reduce the loss function, which is presented as:$$E=\frac{1}{N}\sum_{m=1}^{N}{\left|\right|q_{m}-y_{m}\left|\right|}^{2},$$
where *N* is the total number of iterations, $q_{m}$ is the expected value and $q_{m}$ is the desired value.

#### 4.2.2. CNN

CNN is an efficient neural network model and learns complex features with the help of convolution, non linear activation, drop out and pooling layers [49]. It was designed for image-related tasks such as image segmentation and image classification. In CNN, training is performed in end to end fashion which makes it more efficient. To encode semantic information fully connected layers are utilized at the end of the model. It is a feed forward network where filters are applied to the output of the previous layer to map features. The main components of the CNN model consists of convolutional layers, pooling or sub-sampling layers, flatten layer, activation function, drop out and fully connected layer. Features are extracted by convolutional layers and then the output of the convolutional layers is fed to the fully connected layers. The pooling layer reduces the features mapped by convolutional layers to reduce overfitting. Pooling can be max or average, where the max-pooling layer chooses sharp features as compared to the average pooling layer. The flattened layer converts the data into an array so that it can be fed to the fully connected layer. We utilized the Rectified Linear Unit (ReLU) as an activation function.
(1)y=max(0,i),
where *y* represents the activation output and *i* represents the input. Convolution layers extract local and high level features by assigning weights to kernels during the training phase. CNN has been widely used in disease diagnosis. In binary classification, cross-entropy error is used as a loss function, which has been used in this study. It is computed as shown in Equation (Equation 2).
(2)crossEntropy=−(ilog(p)+(1−i)log(1−p)),
where *i* represents the indicator of class labels, a log is a natural log and *p* represents the probability that is predicted. As CNN is a modification of the backpropagation algorithm, therefore sigmoid is utilized as the error function for output. The CNN model generates output as two neurons for each case of the target class. For the deceased patient, the output will be 1 for the first neuron and 0 for the other neuron. In the case of an alive patient, the values of neurons will be reversed.

CNN has been proved as a robust model for the classification tasks in the medical field. CNN has been utilized by many researchers for various classification tasks such as lung-disease classification [50], segmentation of brain tumors [51] and chest x-rays [52]. In previous literature, CNN has also been analyzed for text categorization such as text sentiment analysis [53], text summarization [54] and text report classification [55]. CNN has been employed to detect vision threatening eye diseases using medical reports in [56].

### 4.3. RNN

Recurrent Neural Network (RNN) is a sequential deep neural network model. It retains the state of the input sequence during the processing of the next sequence. RNN considers the weighted sequence of words in a sentence. It keeps the past knowledge by applying a special loop structure. It is designed to handle text especially to predict the next coming words in a sentence.

### 4.4. LSTM

Long Short Term Memory (LSTM) is an advanced form of RNN and works more efficiently for long term sequences. RNN was facing a vanishing gradient problem that was solved by LSTM. It memorizes specific patterns and performs better than RNN. LSTM consists of three gates—input gate, output gate, and forget gate. It processes word sequences as presented in Equations (Equation 3)–(Equation 5).
(3)$$i_{t}=\sigma \left(x_{t}U^{i}+h_{t-1}W^{i}+b_{i}\right)$$
(4)$$o_{t}=\sigma \left(x_{t}U^{o}+h_{t-1}W^{o}+b_{o}\right)$$
(5)$$f_{t}=\sigma \left(x_{t}U^{f}+h_{t-1}W^{f}+b_{f}\right),$$
where $x_{t}$ is the input sequence, $h_{t-1}$ is the preceding hidden state at current step *t*, $i_{t}$ is the input gate, $o_{t}$ is the output gate and $f_{t}$ is the forget gate.

## 5. Experimental Design

In this section, experimental design and evaluation parameters used to evaluate models in experiments for heart patients’ survival prediction are discussed.

### 5.1. Experimental Details

In this study, a simple yet efficient deep neural MLP model and a powerful deep neural CNN model are utilized to predict a heart patient’s survival. MLP has a dense architecture with an input layer, an output layer and six hidden layers using sigmoid function for activation, comprising of a total eight layer model. We have also conducted experiments with other layers’ structures but the best results were obtained by selecting six hidden layers. The input is in the combination of 13 clinical features of the heart failure clinical record dataset. Hyper parameters has been tuned as; batch size equal to 256, learning rate equal to 0.01, drop out equal to 0.2 and epochs equal to 25. Binary cross entropy has been used as loss function.

As another deep learning model, CNN is implemented for performance evaluation. The CNN architecture design utilized in the experiment of this study is presented in Figure 3. The proposed CNN architecture consists of total 8 layers, 2 convolutional layers, 0.5 dropout value, 2 max-pooling layers and 2 dense layers as presented in Figure 4. The Rectified Linear Unit (ReLU) is utilized as an activation function with CNN. Hyper parameters of the proposed CNN model are presented in Table 3.

### 5.2. Evaluation Measures

Different evaluation measures are being used by researchers to evaluate classification tasks. In analytical research, various performance evaluation tools are playing a significant role in the development process [57]. In this study, accuracy, recall, precision and F-score have been used to evaluate deep learning models. A confusion matrix [58] is used to evaluate these measures. Performance measures for deep learning models are presented in Table 4.

## 6. Results & Discussions

In order to validate the performance of deep learning models for heart failure survival classification, the data were split into training and test sets in a 70:30 ratio. Four deep learning models that are MLP, CNN, RNN and LSTM have been applied to predict heart failure survival. The same train-test ratio and the same feature set have been used by each model. The results of deep learning models have been compared with ML-based models from the literature.

### 6.1. Results

All the experiments are carried out on a 2 GB Dell PowerEdge T430 graphical processing unit on 2x Intel Xeon 8 Cores 2.4 Ghz machine which is equipped with 32 GB DDR4 Random Access Memory (RAM). Deep learning models were implemented using Keras and Tensorflow. The training takes 1 h to give the final result on the heart failure clinical records dataset. Empirical results show superiority of CNN neural network with 0.9289 value of accuracy, 0.94 value of precision, recall and F-score for the classification heart failure survival. Table 5 presents the classification results of deep neural network models in terms of Accuracy, Precision, Recall and F-score. The test results achieved through deep learning models have been compared and it can be observed that all four deep learning models achieved promising results in classifying heart failure survivors. LSTM has shown better results as compared to RNN and achieved a 0.9169 value of accuracy, 0.92 value of precision, recall and F-score. MLP has shown better results in comparison with RNN and LSTM. Furthermore, deep learning models are generally considered as intense with high computational cost. To deal with this issue, deep learning models used in this experiment are simple, efficient and having low computational time. Therefore these deep learning model has potential to be used in real-time heart failure survival classification.

#### 6.1.1. Comparison with ML-Based Models

Mostly ML-based models, such as Decision Tree (DT), Adaptive Boosting Classifier (AdBoost), Stochastic Gradient Descent Classifier (SGD), Random Forest (RF), Gradient Boosting classifier (GBM), Extra Tree Classifier (ETC), Logistic Regression (LR), Gaussian Naive Bayes (GNB) and Support Vector Machine (SVM), give better results while using manual feature engineering. Therefore, we compared the results of deep learning models used in this study with ML-based models used for the classification of heart failure survival in [16]. The authors of [16] performed experiments in different scenarios such as all sets of features without using SMOTE, all sets of features using SMOTE and nine significant selected features with SMOTE for the classification of heart failure survival. As the heart failure clinical record dataset is highly imbalanced, SMOTE was used to make it balanced for the better training of ML-based models.

Table 5 gives the comparison of deep learning models with the best performing ML-based models [16] using SMOTE and without using SMOTE. It has been noticed clearly that CNN neural network model performed better than best performing ML-based model which is RF without using SMOTE and ETC with using SMOTE. Table 6 gives the comparison of deep learning models with ML-based models [16] using feature selection and SMOTE. Deep learning models are showing good results without applying extra techniques such as SMOTE or any feature selection method. It is very significant to notice that the same training and test set is used by each algorithm.

#### 6.1.2. Comparison with Deep Transfer Learning Models

Two transfer learning models, VGG16 and AlexNet, are selected for comparison purposes. AlexNet is a deep neural network model and a winner of the ILSVRC (ImageNet Large Scale Visual Recognition Competition) Competition in 2012 [49]. It has millions of parameters and almost 650,000 neurons are required to train to perform classification. It uses ReLu as a non-linear activation function that assists the model in fast training as compared to the sigmoid function. VGG16 is a 16-layered deep learning-based neural network model and won the ILSVRC-2014 [59]. Instead of millions of parameters, this model uses convolutional layers, padding, max-pooling layer, fully connected layer.

Table 7 presents the comparison of the proposed model with VGG16 and ALexNet. VGG16 has shown better results as compared to AlexNet. The precision, recall and f1-score of the proposed model are slightly lower as compared to VGG16 and better than AlexNet. As the VGG16 model has shown better results in terms of three evaluation metrics but we have to consider the complexity of the models and require more training time to train millions of parameters. Table 8 presents the training time of transfer learning models and the proposed model. With respect to accuracy and training time, the proposed model can be declared as a better performing model for heart failure survival prediction.

#### 6.1.3. Validation of the Proposed Model

To corroborate the generalizability and robustness of the proposed CNN model, 10-fold cross-validation has been conducted on a heart-failure-clinical-records dataset containing records of 299 patients. Table 9 presents the outcome of the proposed CNN deep neural model after performing 10-fold cross-validation. It is proved that the proposed CNN model is robust and can classify heart failure patients into alive or deceased with 0.9262 value of accuracy, 0.9281 value of precision, 0.9399 value of recall and 0.9340 value of F-score.

### 6.2. Discussions

Recently, deep learning models have proven to be very successful for different applications. In this study it has been used for heart failure survival prediction. ML-based models have been applied for the said task and have shown outstanding results with some extra effort such as after dealing with the class imbalance problem and feature selection technique.

MLP have also shown accuracy results with a 0.9201 value which is better than the best performing ML-based model without applying SMOTE which is 0.8889 by RF but slightly lower than the ML-based model with SMOTE which is 0.9262 by ETC as shown in Table 5. Precision and F-score achieved by MLP are equal to that of ML-based models as shown in Table 5. However, MLP is exhibiting reasonable results for the classification task of an imbalanced dataset without applying the extra effort of manual feature engineering.

Mostly, CNN shows superiority in performance among other neural network models. It works with the raw data and explores spatial relationships among data by extracting its own feature set by assigning different weights with the help of kernel. In this study, 13 features from the heart failure clinical record dataset are fed to the embedding layer as input and hence provide a better classification result.

To report the overall performance of the deep learning model, Figure 5 provides the training and testing accuracy curve which proves that the model is not overfitting. It can be concluded that CNN performs better as compared to MLP and other ML-based models used in the literature. To prove the robustness of the proposed model, we performed (k-fold) cross validation by setting the value of k to equal 10. Empirical results proved that CNN based proposed model is stable and achieved 0.9263 value of accuracy.

We have seen in the discussion section that the proposed learning model is outperforming all other approaches, either machine or deep learning, based on giving the highest accuracy, precision, recall, and F-score. The stability of the proposed model is also tested using 10-Fold cross-validation. For a fair comparison between the related work and the proposed model in terms of technicality and novelty, we have designed Table 10. From Table 10, we can observe that there is no such research work that utilizes more than five features to transfer data to a cloud server and do the computation on all. Furthermore, none of the mentioned research work stores patient data over the months. Their data-storage capacity is between 24 and 48 h which is less with which to make medical predictions.

To prove the effectiveness and robustness of the proposed model, a comparative analysis has been performed with the existing studies from the literature on the same dataset. Table 11 presents the comparison of the proposed approach with the existing literature. Results reveal that the proposed model proves its superiority over other existing approaches with a 0.9289 value of accuracy.

There are some limitations of this study that need to be considered. First, the dataset used in this study is of only 299 patients, which is very small. There is a need for a large dataset for generalization, better training and for improvement in the performance of deep learning models. Second, the data are highly imbalanced, so the model cannot be trained well on a minority class. It would be advantageous to use a large-sized and balanced dataset and results could also be improved in this way.

For future work, our aim is to collect large-sized datasets to improve the performance of the models for practical use. In addition, more deep learning models will be explored for better results.

## 7. Conclusions

A smart healthcare framework is proposed that integrates IoT and cloud technologies for heart failure patients’ real-time monitoring. The framework uses IoT-based sensors to acquire heart patients’ states. These signals are then transmitted to the cloud unit for further processing. The deep learning model detects the patient’s survival using signals. Improving the prognosis of heart disease patient mortality by processing health-related symptoms using deep learning algorithms will assist in rescuing the lives of patients with acute heart failure. By investigating factors leading to acute heart failure, a patient’s survival rate can be increased by following preventive measures. In this work, an effective, efficient, and simple smart healthcare framework is proposed for the monitoring of patients with heart failure.

To prove the effectiveness of the model, the deep learning model is trained on a heart failure clinical record dataset. Thirteen features from the heart failure clinical record dataset were utilized as input to the deep neural network. This is the first reported heart failure survival prediction task based on deep learning models. The proposed CNN-based deep learning model did not use any extra effort to handle the dataset before training such as making the dataset set balanced and feature selection techniques which make it more appropriate to be used in real-time heart failure survival classification. Two deep learning models, MLP and CNN, were trained on the same dataset. Results reveal that CNN showed robust results after training on raw data for heart failure survival classification.

Comparison results of the proposed CNN model show its superiority over the ML-based model applied on the same dataset in previous literature [16]. To prove the generalizability of the proposed CNN model, 10-fold cross-validation has been performed which also proves its robustness and performance stability by showing a 0.9263 value of accuracy. It can be concluded that characteristics like easy implementation and low complexity make CNN feasible for heart survival classification. Then these results are provided to the medical experts so that follow-up of patients can be performed. The future aim is to explore more deep learning models for the proposed smart framework.

## Figures and Tables

**Figure 1 sensors-22-02431-f001:**
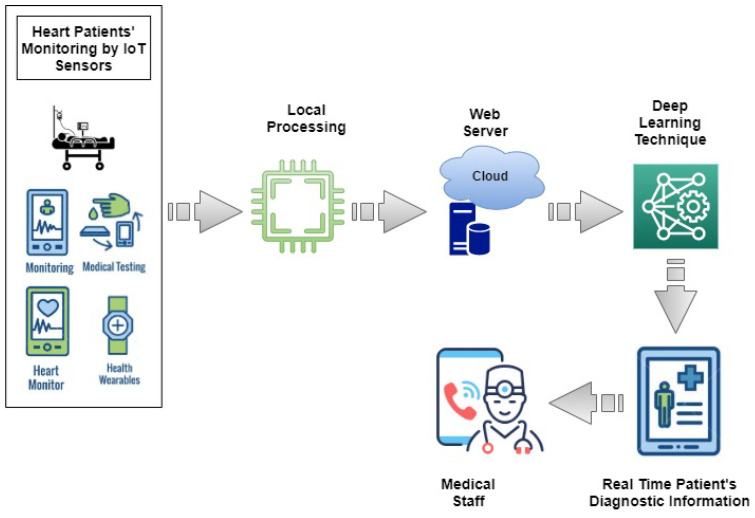
Smart Healthcare Framework to monitor heart-failure patients.

**Figure 2 sensors-22-02431-f002:**
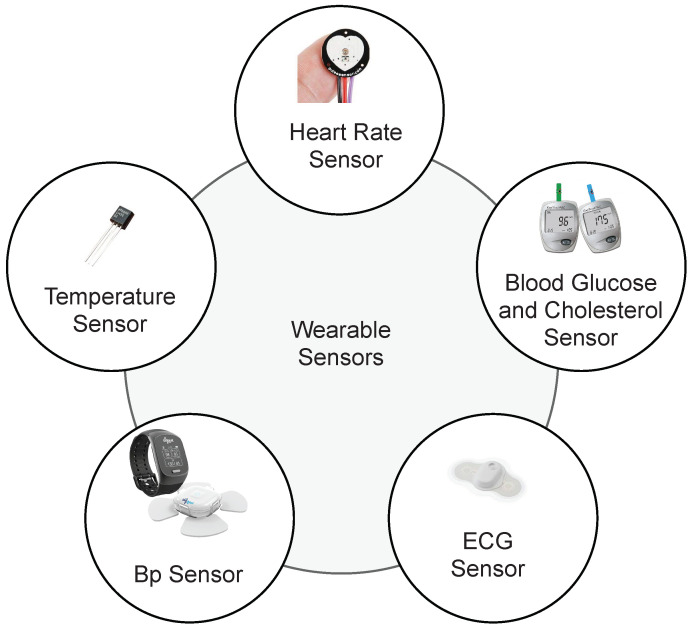
IoT-based Sensors to monitor patient health.

**Figure 3 sensors-22-02431-f003:**
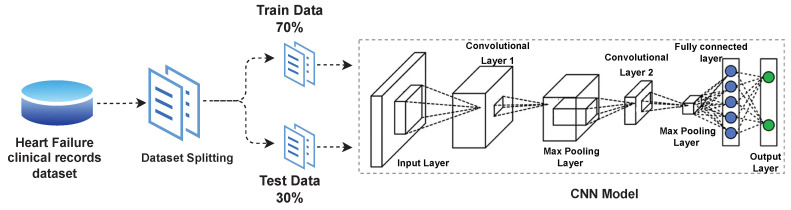
CNN-based Deep Learning Architecture.

**Figure 4 sensors-22-02431-f004:**
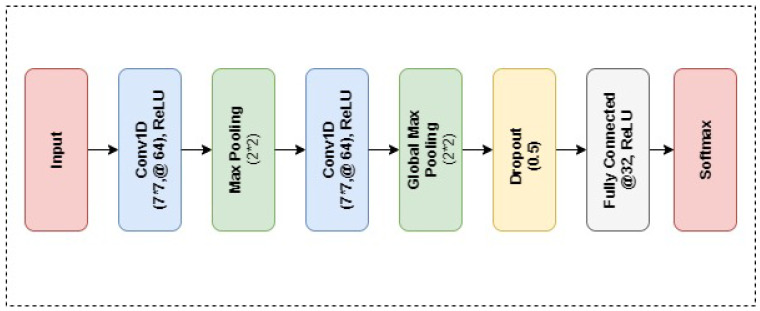
Layered Architecture of the Proposed CNN neural network.

**Figure 5 sensors-22-02431-f005:**
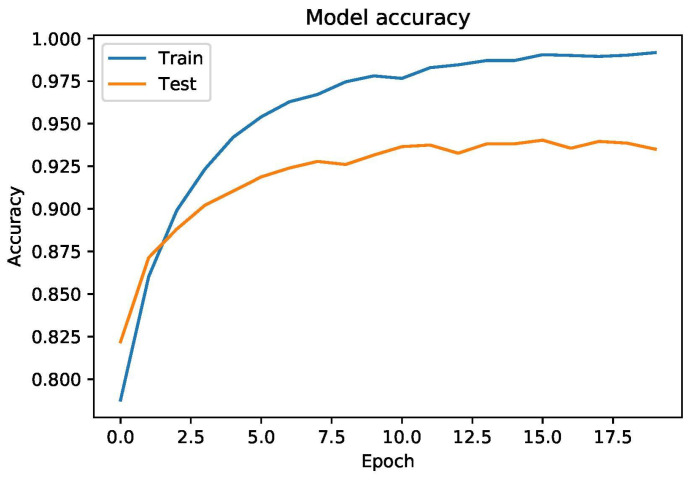
Training and Testing accuracy on heart failure-clinical-record dataset.

**Table 1 sensors-22-02431-t001:** Overview the existing studies from the literature.

Reference	Method	Data	Findings	Limitation
[24]	Neural Network (NN), SVM, Fuzzy Genetic, CART and Random Forest	Database of heart failure patients 136 records from 90 patients,	CART proved as most appropriate in evaluating heart failure severity and its type	Proposed model did not generalize well due to small sample size. Accuracy result is quite low in severity assessment.
[28]	Naïve Bayes, KNN, Decision Tree, and Random Forest	Heart disease patient dataset consisting 303 instances obtained from UCI	This paper find the chance of heart disease in patients.	The authors considered 14 attributes out of 76 attributes and results could be improved by applying feature selection.
[29]	Naïve Bayes, KNN, Decision Tree, Random Forest and HRFLM model(combination of Random Forest and Linear Method)	Four datasets (Cleveland, Hungary, Switzerland, and the VA Long Beach)	The proposed hybrid model predicted heart disease better than machine learning models	More combination of models along with feature selection need to be explored.
[31]	Multiple Kernel Learning and Adaptive Neuro Fuzzy Inference System	KEGG Metabolic Reaction Network dataset.	Experiments have been applied in a two-fold approach in classifying patients into heart disease and healthy ones.	Very small number of features or parameters are considered in the experiment
[37]	Fisher ranking method, generalized discriminant analysis(GDA)	NSR-CAD and $Self_{N}$SR-CAD	Authors proposed noninvasive approach using (GDA) to automatically detect coronary artery disease using heart rate variability signals	Models need to train on a large dataset of heart rate variability signals for generalizability.
[43]	Ensemble of Random Forest, Gradient Boosting Machine, and XG Boost.	Z-Alizadeh Sani, Statlog, Cleveland, and Hungarian datasets	Used an ensemble model to detect coronary artery disease.	A stacked ensemble of three models also increased the complexity and the cost of the model.
[44]	Ensemble of BiLSTM, BiGRU and CNN	heart disease dataset	Ensemble learning framework using deep model was applied to deal with the problem of an imbalanced heart disease dataset.	The proposed approach has not tested on a benchmark dataset.

**Table 2 sensors-22-02431-t002:** Dataset Specifications.

Sr No.	Attributes	Description	Range	Measured In
1	Time	Followup period	4–285	Days
2	Event (target)	If the patient died in the followup time	0,1	Boolean
3	Gender	Man or woman	0,1	Binary
4	Smoking	If the patient smokes	0,1	Boolean
5	Diabetics	If the patient has diabetics	0,1	Boolean
6	B.P	If the patient has blood pressure issue	0,1	Boolean
7	Anaemia	Decrease in red blood cell or haemoglobin	0,1	Boolean
8	Age	Age of the patient	40–95	Years
9	Ejection fraction	Percentage of blood leaving the heart at each concentration	14–80	Percentage
10	Sodium	Level of sodium in the blood	114–148	mEq/L
11	Creatinine	Level of creatinine in the blood	0.50–9.40	mg/dL
12	Platelets	Platelets in blood	25.01–850.00	kiloplatelets/mL
13	CPK (creatinine Phospho)	Level of CPK enzyme in the blood	23-7861	Mcg/L

**Table 3 sensors-22-02431-t003:** Summary of hyper-parameter values for CNN.

Parameter	Value
Embedding dimension	300
Batch size	256
Pooling	2 × 2
No. of filters	5 × 64
Max_Sequence_length	130
Epochs	25
Optimizer	Adam
Function	Binary cross entropy

**Table 4 sensors-22-02431-t004:** Performance Evaluation Parameters.

1	**Accuracy =** $\frac{\mathrm{Number}\;\mathrm{of}\;\mathrm{correctly}\;\mathrm{classified}\;\mathrm{predictions}}{\mathrm{Total}\;\mathrm{predictions}}$
2	**Precision =** $\frac{TP}{TP+FP}$
3	**Recall =** $\frac{TP}{TP+FN}$
4	**F-Score =** $2*\frac{precision.recall}{precision+recall}$

TP = True Positive, TN = True Negative, FP = False Positive, FN = False Negative.

**Table 5 sensors-22-02431-t005:** Performance comparison of the proposed CNN deep neural network models with RF & ETC [16].

Model	Accuracy	Precision	Recall	F-Score
CNN	0.9289	0.94	0.94	0.94
MLP	0.9201	0.93	0.92	0.93
RNN	0.9001	0.88	0.90	0.89
LSTM	0.9169	0.92	0.92	0.92
RF without SMOTE [16]	0.8889	0.89	0.89	0.89
ETC with SMOTE [16]	0.9262	0.93	0.93	0.93

**Table 6 sensors-22-02431-t006:** Accuracy comparison of deep learning models with machine learning models using significant features after applying SMOTE.

Models	Accuracy
DT [16]	0.8778
AdaBoost [16]	0.8852
LR [16]	0.8442
SGD [16]	0.5491
RF [16]	0.9188
GBM [16]	0.8852
ETC [16]	0.9262
GNB [16]	0.7540
SVM [16]	0.7622
RNN	0.9001
LSTM	0.9169
MLP	0.9201
CNN	0.9289

**Table 7 sensors-22-02431-t007:** Performance comparison of the proposed CNN deep neural network models with Transfer learning models.

Model	Accuracy	Precision	Recall	F-Score
VGG16	0.9129	0.90	0.92	0.91
AlexNet	0.9071	0.90	0.90	0.90
CNN	0.9289	0.94	0.94	0.94

**Table 8 sensors-22-02431-t008:** Statistics for required training time for classifiers.

Model	Training Time
Proposed approach	35 min
VGG16	39 min
AlexNet	47 min

**Table 9 sensors-22-02431-t009:** Ten fold cross-validation results using CNN deep model.

Fold Number	Accuracy	Precision	Recall	F-Score
1st-Fold	0.915	0.916	0.921	0.911
2nd-Fold	0.912	0.907	0.922	0.926
3rd-Fold	0.911	0.923	0.923	0.934
4th-Fold	0.918	0.907	0.949	0.935
5th-Fold	0.904	0.911	0.948	0.933
6th-Fold	0.916	0.926	0.947	0.932
7th-Fold	0.924	0.907	0.916	0.941
8th-Fold	0.914	0.914	0.945	0.937
9th-Fold	0.902	0.924	0.954	0.918
10th-Fold	0.947	0.945	0.957	0.949
**Average**	**0.9263**	**0.9281**	**0.9399**	**0.9340**

**Table 10 sensors-22-02431-t010:** Comparison between the related work and the proposed model.

Work	Features > 5	Monthly Patient Record	AI	IoT
[60]	×	×	✓	✓
[61]	×	×	✓	✓
[62]	×	×	✓	✓
[63]	×	×	✓	✓
[64]	×	×	✓	✓
[65]	×	×	×	✓
[66]	×	×	×	✓
[67]	×	×	×	✓
**Proposed Model**	✓	✓	✓	✓

**Table 11 sensors-22-02431-t011:** Comparison of the proposed approach with the existing literature.

Authors	Models	Accuracy
Parthiban et al. [25]	Naïve Bayes	0.74
Kumar Dwivedi [68]	Logistic regression	0.85
Vembandasamy et al. [69]	Naïve Bayes	0.86
Shah et al. [28]	K-NN	0.90
**Proposed Model**	**CNN**	**0.9289**

## Data Availability

The datasets generated during and/or analysed during the current study are publicly available. The link of the dataset is shared in the Section 4.1 of the manuscript. The dataset and code is available from the corresponding author on reasonable request.

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
