# Peer review of "IoT Based Smart Monitoring of Patients’ with Acute Heart Failure"

_sensors, 2022, doi:10.3390/s22072431_

Round 1
Reviewer 1 Report
The paper is now more appropriate for pubblication.
Author Response
We would like to extend our sincere gratitude to the anonymous reviewer for his/her invaluable and timely feedback. We have modified our manuscript according to the suggestions and comments we received. All these changes have been fully addressed in this Response Letter and have been reflected in the manuscript as well.
Reviewer 2 Report
Authors have improved manuscript at small extent, still there are major changes to be made
Major Reviews for Authors
- CNN model section in Fig.3 is blurred and low quality, authors are highly recommended to redraw it with high visibility
- Results section is weak and not convincing, so more results must be added for the betterment of the reading for readers
- Flowchart and pueudocode are missing in section 4, which are very important to find the significance / novelty of the research
- There are many auhtors in the paper, but contribution/novelty is not too much, so authors are highly recommended to improve the novelty section
- Comparison with existing state of the art frameworks is missing, authors can hight with more latest and relevant works, what is uniqueness of their work ? because there are several works on the frameworks of 'Acute patient's monitoring and CNN models'
- Title should be bit revised, because it is not revealing something new to motivates the readers
Paper needs major changes to be fixed
Author Response
Dear reviewer, we have added response letter file which address all of your concerns. Thanks for your valuable comments.

Round 2
Reviewer 2 Report
Authors have improved the paper but still there are some recommendations
- Results section is weak and not-convincing, so authors are highly suggested to get more results to justify their idea
- Comparision with exisiting works is missing, which is very important to include
Author Response
Dear Reviewer, we have tried our best to address your all concerns and meet the final acceptance this time. We have also modified/updated the entire manuscript to improve the clarity and readability of the manuscript.

This manuscript is a resubmission of an earlier submission. The following is a list of the peer review reports and author responses from that submission.
Round 1
Reviewer 1 Report
The paper presents IoT based smart healthcare system for acute heart failure monitoring. The paper’spaper’s idea sounds good, but the paper has many serious flaws. First, it seems like the authors only combined existing solutions for acute heart failure monitoring.
Why the proposed system of patient monitoring? What is the difference and advantage of the proposed system for acute heart failure monitoring rather than all others?
What are the performance metrics used for the IoT part of the proposal? The details of IoT server development and deployment is missing?
How the deep learning models are deployed in the IoT environment, please explain objectively.
The English and formatting style of the paper needs significant improvement.
Update the Data Availability Statement: The datasets used in the study is already publicly available, and this statement contradicts your description.
Abstract: Dataset used in this study contains
13 13 features and was attained from the UCI repository known as Heart Failure Clinical Records.
““Eventually, the same dataset was
explored by [16] using the full set of features in comparing ML-based models to predict heart failure survival. The authors also applied Synthetic Minority Oversampling Technique (SMOTE) to make the dataset balanced.”” at line 77-- Revised all text where authors are referring to their previous work. First, make a background based on your existing study and then what is your new proposal.
““Major contributions of this study are:
Designed an efficient smart healthcare system using IoT and cloud based technologies to facilitate medical professionals in monitoring patients with heart-failure.””
(Please revise the contribution section and present what is the novelty of the work?
for example, see these studies : https://link.springer.com/content/pdf/10.1007/s42979-020-00195-y.pdf
https://ieeexplore.ieee.org/abstract/document/8529141
Please explain in detail your cloud IoT architecture and present it through a figure. For instance, read this study :
https://www.mdpi.com/2073-8994/13/2/357
What is pathology report data? The UCI dataset or its another dataset, please explain academically.
““The performance of deep learning models is investigated in predicting the survival of heart patients.””
What is the novelty of the work?
““The performance of the proposed CNN deep learning model is compared with MLP and ML-based algorithms trained on same dataset.””
This comparison is not significant first, if authors are using datasets which other studies have used, then the comparison should be made with existing studies on the same dataset from UCI. Secondly, The CNN and other algorithms used are already used by studies? What is the contribution? To justify, please add comparision section by comparing your study with existing studies from the literature.
Please revised the introduction section thoroughly, there seems to be heaped text, and the paragraphs do not have coherence.
Please revise the related work section with a summary of existing work in table form, summarizing the advantages and disadvantages of the existing study. Please also consider recent literature from the web of science journals and revised the related work section.
Figure 1, and 2 should be revised to be more specific to the proposal, it seems generic diagrams from the internet. Please revised the materials and methods sections. The content is from existing literature, cite and reference the work. The better place for this is the introduction and related work section, i.e 4.2 section.
The Conflicts of Interest: and author’sauthor’s contributions statements contradict each one.
For instance, if ““The funding agency had no role in the design of the study; in the collection, analyses, or interpretation of data; in the writing of the manuscript, or in the decision to publish the results””.
Then what is the contribution of the Funding acquisition, Michele NAPPI;”” please explain the investigations done in the context of this study.
"A smart healthcare framework is proposed that integrates IoT and cloud technologies for heart failure patient’s real-time monitoring."
Explain smart healthcare in more detail.
Would you please provide a real-life case study of the study, or at least the simulation results when the proposed model is deployed in cloud IoT architecture? what is its impact on the following performance measures:
latency, reliability, failure rate.
Figure 4. Performance comparison of deep learning models.
What are deep learning models, MLP and CNN?
6.2. Discussions
"Recently deep learning models proven to be very successful for different applications, but to the best of our knowledge, this is the first time it has been used for heart failure survival. "
Please revise the statement, and consider recent studies by searching using google scholar, the keyword should be "heart failure survival deep learning"
After thoroughly revising the manuscript, the results and summary of the new findings should be updated in the abstract and conclusion.
Author Response
We have attached the response to reviewer file.

Reviewer 2 Report
The work presented in your paper describes a system, based on the IoT and artificial intelligence (AI) methods to monitor the health state of patients suffering acute heart failure. This subject is timely and plein of actuality since intelligent healthcare systems are more and more present in our daily lives, and their use will be increased in next years, together with new technologies like IoT and AI. However, in its current state, your work includes a number of weaknesses and thus, in my opinion, it cannot be accepted for publication in this journal. Next, you will find some comments with the aim of helping you to enhance your article.
Even if you have summarized between lines 96 and 103 the major contributions of your paper, they are not clearly supported by the information provided in the text. Indeed, in the Introduction and Related Work sections, you give some references, but it is not easy to identify the differences between these works and yours. It is necessary to analyze exhaustively the advantages and disadvantages of each one of the papers in reference and compare them with your work. To carry out this analysis, you should define the comparison criteria, and the importance of each one of these criteria. The results of this analysis can be presented in a table, in order to make them comprehensible for readers.
In section 3, you present an overview of the Smart Healthcare framework, which is in addition illustrated in Figure 1. Nevertheless, there is no information concerning the employed sensors to monitor patients, the local processing procedure (SW and HW), how the diagnosis is obtained in real time, how the diagnosis is sent and presented to the medical staff.
In section 4, Table 1 provides the dataset specifications, that have been employed in your neural network. However, there is no information regarding the weight of each attribute, because I suppose that they have not the same degree of importance. You also present the general equations of MLP and CNN, but the definition and the particularities of you CNN are not clearly visible in your paper (how does it work, the training procedure, etc.).
Section 6 discusses the obtained results, and compares them with those got with the SMOTE procedure. Why to compare your results with SMOTE? This choice must be better justified. It is also necessary to compare your results with other methods, to correctly validate the advantages of your solution.
Concerning the references, please verify their correct numbering. As an example, I think that references 18 and 19 are not well numbered according to the references list.
As final remarks, your work is based on an existing database, which is a good way to obtain real data for your CNN. Nonetheless, it is crucial to explain how this dataset has been obtained, to correctly demonstrate its adequation to be employed as a support for your work. Finally, the title of the paper should be modified, since you mainly speak about the AI method, but there is no information regarding the IoT system: sensors, embedded electronics, communication technologies, protocols, to point up a number, which are at the heart of any IoT system.
Author Response

(The authors gave the same response as above.)

Reviewer 3 Report
The paper presents a Smart Healthcare system to predict heart failure using deep learning algorithm.
The authors have to explain why they decided to use MLP and CNN algorithms also in respect to the state of the art.
Which sensors are used in the work to measure the data? In chapter 3 wearables and fixed sensors are mentioned but without technical explanation. Since the journal is Sensors and the title of the article is IoT-based Healthcare system, the authors have to deeply explain the measurement setup of this study, including sensors and devices, technical information (sampling frequency, accuracy, etc.), etc. The information regarding the platform are needed.
In Figure 1 the authors mentioned a local processing. How it was done?
Figure 3 is partially explained in the text as Table 2.
How the authors applied the SMOTE method at the dataset?
Figure 4 is not useful since the same results are already shown in Table 4.
Author Response
We have attached the response to the reviewer file.

Reviewer 4 Report
Authors have highlighted the emerging and core issue, but still there are major issues to be fixed.
Reviews to Authors
- Title must be simple, clearer and nicer.
- Spell out each acronym the first time used in the body of the paper. Spell out acronyms in the Abstract by extending it.
- The abstract can be rewritten to be more meaningful. The authors should add more details about their final results in the abstract. Abstract should clarify what is exactly proposed (the technical contribution) and how the proposed approach is validated.
- What is the motivation of the proposed work?
- Introduction needs to explain the main contributions of the work clearer.
- The novelty of this paper is not clear. The difference between present work and previous Works should be highlighted.
- Authors must explain in detail the introduction section.
- Authors must develop the framework/architecture of the proposed methods
- There is need of flowchart and pseudocode of the proposed techniques
- Proposed methods should be compared with the state-of-the-art existing techniques
- Research gaps, objectives of the proposed work should be clearly justified.
To improve the Related Work and Introduction sections authors are highly recommended to consider these high quality research works <‘ Power Management Strategies for Medical Information Transmission in Wireless Body Sensor Networks’, IEEE Consumer Electronics Magzine, Vol.9, No.2, pp.47-51, 2020>, < A Novel Adaptive Battery-Aware Algorithm for Data Transmission in IoT-Based Healthcare Applications, Electronics, MDPI, vol.10, no.4, pp.367, 2021>
- English must be revised throughout the manuscript.
- Limitations and Highlights of the proposed methods must be addressed properly
- Experimental results are not convincing, so authors must give more results to justify their proposal
- Figures quality and visibility are poor, so authors are strongly suggested to redraw the all figs with more clarity and high resolution
Finally, paper needs major improvements
Author Response

(The authors gave the same response as above.)

Round 2
Reviewer 3 Report
The paper improved after revision but I suggest another time to the authors to explain the source of the data. In particular, in research papers, you must to explain deeply the work to provide to the scientific communities the possibility to fully understand your work. In this case and since the journal is Sensors, you must explain the sensors (e.g. wearables and fixed sensors) that you used and cited in the paper. You can have a look to this previous paper published some years ago on Sensors that explain in a synthetic and explanatory way the sensors used in that work.
Monteriù, A., Prist, M. R., Frontoni, E., Longhi, S., Pietroni, F., Casaccia, S., ... & Revel, G. M. (2018). A smart sensing architecture for domestic monitoring: Methodological approach and experimental validation. Sensors, 18(7), 2310.
Author Response
Respected reviewer, we have addressed your concern in the PDF file attached to this response submission. I hope we will meet final acceptance after that. Thanks for your kind guidance in improving the quality of the manuscript.

Reviewer 4 Report
Authors have improved the paper, but still major improvements are needed in the important sections
Comments
- Results section is very weak, so authors are highly encouraged to get and add sufficient results
- To improve the Introduction and Related work section it is highly recommended to add and read high quality works , Information Fusion, Elsevier, Vol. 53, No.2020, pp.155-164, 2020>
- It is suggested to put more information the captions of all figures for the better understanding and clarity
- Authors are strongly recommended to add sub-section in the Results section to give some details about the tools, datbases, and datsets were considered in their research
Paper cannot be considered in its current form so needs major revisions
Author Response
Respected reviewer, we have addressed your all concerns in the PDF file attached to this response submission. I hope we will meet final acceptance after that. Thanks for your kind guidance in improving the quality of the manuscript.
